# T-Racing: a modern tool for supporting epidemiological investigation in animal disease outbreaks in Italy

Luca Candeloro[1], Lara Savini[1]*, Annamaria Conte[1], Paolo Calistri[1], Diana Palma[1], Samuel Perticara[1], Anna Cecília Trolesi Reis Borges Costa[2], Alessio di Lorenzo[1], Susanna Tora[1], Sara Serrani[1], Michela Toro[1], Karina Diniz Baumgarten[2,3], Fabrizio De Massis[1]

**1** Istituto Zooprofilattico Sperimentale dell'Abruzzo e del Molise "G. Caporale", Italy, **2** Departamento De Medicina Veterinária, Escola de Zootecnia e Medicina Veterinária, Universidade Federal de Lavras, Lavras Minas Gerais, Brazil, **3** State Department of Animal Health, Companhia Integrada de Desenvolvimento Agrícola de Santa Catarina, CIDASC, Florianópolis, South Carolina, Brazil

* l.savini@izs.it

## Abstract

The One Health approach unites efforts across human-animal-environment interfaces against shared threats like zoonotic diseases. T-Racing is a Shiny web application, that supports epidemiological investigations and helps contain livestock-related disease spread, aligning with multidisciplinary principles to safeguard public health. The application uses Temporal Network Analysis techniques to address the dynamic nature of animal trade, facilitating backward and forward tracing strategies. T-Racing leverages web services to retrieve data from multiple sources simultaneously and in near real-time through the plumber package and is distributed using Shinyproxy. T-Racing manages and analyze extensive and diverse datasets within the same environment, including animal movement data, disease outbreak data, and genomic data, all obtained from Italian National databases. In this work, we show T-Racing's capabilities by simulating epidemiological investigations of brucellosis and tuberculosis outbreaks that occurred in non–endemic areas of Italy. To further highlight its capabilities, an interactive demo of T-Racing is available, showcasing its potential and features. This tool supports epidemiological investigations by adopting a data-driven approach, guiding users through the analysis via an iterative process while leveraging their expertise. Therefore, it enables faster data analysis, improves understanding of disease transmission patterns, and facilitates prompt and targeted interventions.

## 1. Introduction

Preventing and controlling communicable diseases in livestock through early detection of outbreaks increases the effectiveness of the response and reduces the social, economic, and environmental costs associated with these events [1,2]. In Italy, veterinary surveillance programs target several zoonotic diseases in livestock, including brucellosis, tuberculosis, leptospirosis, rabies, salmonellosis, campylobacteriosis, listeriosis, and echinococcosis etc.

**Data availability statement:** All data used are publicly available, a representative sample of these datasets has been anonymized and spatial coordinates have been noised and rounded to the second decimal to be included in the demo version, publicly accessible at the link: https://demo.izs.it/app_direct/tracing/.

**Funding:** This research was funded by Italian Ministry of Health (MSRCTE0219). The funders had no role in study design, data collection and analysis, decision to publish, or preparation of the manuscript.

**Competing interests:** The authors have declared that no competing interests exist.

Animals serve as an excellent proxy for monitoring both novel and known zoonotic pathogens given that more than 60% of emerging infectious diseases in humans originate from animal sources [3–5]. This underscores the critical role of animal health (AH) in implementing of the One Health approach, by detecting, monitoring, and predicting health hazards that could also impact environmental and human health since some pathogens can survive long periods in the environment [6]. Among the diseases that could affect human health are tuberculosis and brucellosis, both chronic infectious diseases caused by bacteria. Bovine tuberculosis, caused by *Mycobacterium bovis* [7], is a zoonotic disease that was estimated to have a human prevalence varying from 0.4% to 76.7% worldwide [8]. Animal brucellosis caused by *Brucella* spp., is a zoonosis of utmost significance, since it was projected to globally affect 2.1 million people by year [9]. The control of both diseases is anchored on the surveillance and control in animals, with the traceability of animal cases being of paramount importance to prevent human infections, especially for human brucellosis that does not have vaccination.

In this context, animal movement is often considered a major risk for disease spreading between herds [10] and to humans when zoonoses are involved. Hence, livestock transportation data is registered in national and international databases to facilitate contact tracing in the event of an outbreak [11–17]. The use of animal movement data facilitates tracing the spread of diseases within animal populations and the potential routes of exposure being critical for effective disease control and crucial in the final steps of eradication programs [18]. Contact tracing plays a pivotal role in identifying the necessary actions to break the transmission chain, to reduce the spread and duration of infectious diseases, thus preventing new cases [19]. When zoonoses are considered, this also means protecting public health (PH), as effective contact tracing subsequently aids in providing assessments to stakeholders, which are useful for quickly identifying and implementing containment, surveillance, and control systems.

Technology plays a fundamental role in developing online tracing applications, that can improve the results of epidemiological investigations. Advanced analytics and link analysis can be used to uncover hidden patterns in the spread of diseases [2,20]. Therewith, tracing applications have been developed to assist AH officials in quickly identifying the animals involved in an outbreak, where and when the infected animals are located, and whether other susceptible animals might have been exposed to the disease. These actions would be taken by uncovering the missing or unexpected links from the contact details, identifying the animals that should undergo testing, tracking the geographical spread of the disease, and determining which farms are at the highest risk. Thus, tracing applications can help in disease control and surveillance during critical outbreak stages, saving time, preventing further spreading, minimizing costs and reducing the consequences of the disease outbreaks. Tracing applications may enable AH services, epidemiologists, and competent authorities to respond quickly by identifying the "hot spots" and implementing effective targeted control and eradication measures [2,21]. Although it may sound simple, the tracing application can become overwhelmingly complex and resource-intensive, however, using data visualization and analytics can significantly improve the performance and effectiveness of epidemiological investigations, making the application a valuable tool to help risk assessors in delivering mitigating options to decision makers. Due to the recent COVID-19 global pandemic, contact tracing strategies have been widely used and many contact-tracing web apps have been developed for this purpose in human health [22–23]. In veterinary epidemiology, there is limited literature on contact tracing web applications, with notable examples being EpiContactTrace [2] and EpiExploreR [24]. EpiContact-Trace, an open-source R package, allows end users to conduct forward and backward contact tracing by analyzing livestock movement data. However, it requires technical knowledge in R programming, manipulation and adjustment of data, which limits its accessibility for many users having different expertise. Additionally, it lacks the ability to integrate heterogeneous and

epidemiologically relevant data necessary for a comprehensive epidemiological investigation. EpiExploreR, an R shiny web application, offers tools for disease mapping, spatiotemporal analysis, and integration of diverse datasets, but its complexity requires specialized expertise making it less accessible for field operators and veterinarians who require immediate and user-friendly contact tracing tools. Moreover, both tools lack effective management of large networks, which is crucial for avoiding chaotic and unusable results.

Even though there are tracing applications already developed in the AH field, there is still a need for a versatile and user-friendly web tool that can manage various and heterogeneous datasets and incorporate analytical features to quickly perform forward and backward tracing of outbreaks within a spatial-temporal environment. Therefore, the aim of this paper was to introduce T-Racing, a web-based operational tool developed to assist epidemiological investigations in Italy. To further support this objective, an interactive demo of T-Racing has been made available to the scientific community, illustrating its potential and features for readers of this article (https://demo.izs.it/app_direct/tracing/). Unlike existing tools that may require specialized knowledge or are limited by complexity, T-Racing is designed to be easily accessible to local and national authorities working in AH and PH, at different levels, from outbreak investigation and risk assessment to decision-making processes. Based on our knowledge, T-Racing offers several distinctive features that overcome the limitations of existing applications. These include its ability to integrate diverse data sources, a user-friendly interface, and scalability. It integrates certified data directly from national databases, ensuring that the underlying data is both reliable and easily interpretable. This is possible because of the tool's foundation in a deep knowledge of the data and the Italian animal Identification and Registration system (I&R). T-Racing allows users to explore and analyze the complex spatiotemporal pathways that shape disease spread enabling them to draw meaningful conclusions based on a data-driven, user-guided approach.

## 2. Materials and methods

### 2.1. Data sources and collection

T-Racing gathers near-real-time data from Italian National Databases which include disease notified outbreaks obtained from the National Information System for the Notification and Management of Animal Diseases (SIMAN) which manages data on more than 100 diverse animal diseases, including both zoonotic and non-zoonotic diseases [25], animal movements records extracted from the Italian National Database for Animal Identification and Registration (NDB) [26], and genomic data, such as Multi Loci Sequence Type (MLST), provided by National Reference Centre for Whole Genome Sequencing of microbial pathogens database and bioinformatics analysis (GenPat) [27,28]. These systems can be accessed through the VETINFO portal (https://www.vetinfo.sanita.it/), which is a secure platform to collect and present data for managing Italy's Animal Health and Food Safety system. VETINFO provides unified access to various stakeholders, including PH institutes, AH services, farmers, and competent authorities. To ensure user trust and regulatory compliance, the portal strictly adheres to General Data Protection Regulation (GDPR) compliance techniques, requiring users to access it with personal credentials and mandating specific functionalities for transparent and secure data reporting. The management of these databases is the responsibility of the Istituto Zooprofilattico Sperimentale dell'Abruzzo e del Molise "G. Caporale" (IZS-Teramo) on behalf of the Italian Ministry of Health.

T-Racing collects detailed data for each reported suspected or confirmed outbreak, including the disease name (i.e., brucellosis, tuberculosis, leptospirosis, rabies, salmonellosis, campylobacteriosis, listeriosis, foot and mouth disease, african swine fever, blutongue, etc.), unique outbreak identifier code, geographical coordinates (which are related to the outbreak

location that can be a farm or another site where the infected case was identified by the local AH service), region, province, administrative unit, suspected and confirmed outbreak date, the latest outbreak state (i.e., suspect, extinct, and confirmed), and species involved (including animal, vector species such as mosquitoes and flies, and birds, etc.). Animal movement data includes the origin and destination unique holding identifier codes, animal single identification codes or batches of traded animals per species (including horses, cattle, buffalo, pigs, sheep and goats), and the date of the movement. The system also considers various holding attributes (livestock properties) for both origin and destination, such as the holding type (e.g., farm, slaughterhouse, staging point, transit station, market, assembly centre, pasture, foreign country, etc.), production type (e.g., milk, meat, wool, reproduction, etc.), address, and geographical coordinates. Additionally, in terms of genomic data, T-Racing records the holding code, animal single identification code, Sequence Type number (an identifier for classifying bacterial strains, such as Brucella, based on selected genetic loci through MLST), and date of sampling. Furthermore, T-Racing relies on data collected over the past 7 years extracted from the databases through daily scheduled procedures.

## 2.2. Temporal network analysis methods in T-Racing

The application uses Temporal Network Analysis techniques [29] to address the dynamic nature of animal trade, enabling quick and efficient Backward and Forward Contact Tracing strategies (BCT and FCT). In the T-Racing environment, the animal trading network is represented as a time-ordered directed and weighted network, where holdings are the nodes, animal movements between holdings on a given day are the edges, and the number of moved animals is the weight (see Fig 1 on the left).

Animal movements are recorded in the NDB at the administrative farm owner (farmer) level. Consequently, multiple farmers from the same holding may be represented as different nodes (see Fig 1, top right), causing the network to be artificially fragmented. This fragmentation significantly impacts the analysis of diseases affecting multiple animal species. To address this issue, the application allows for network aggregation, ensuring that each node represents

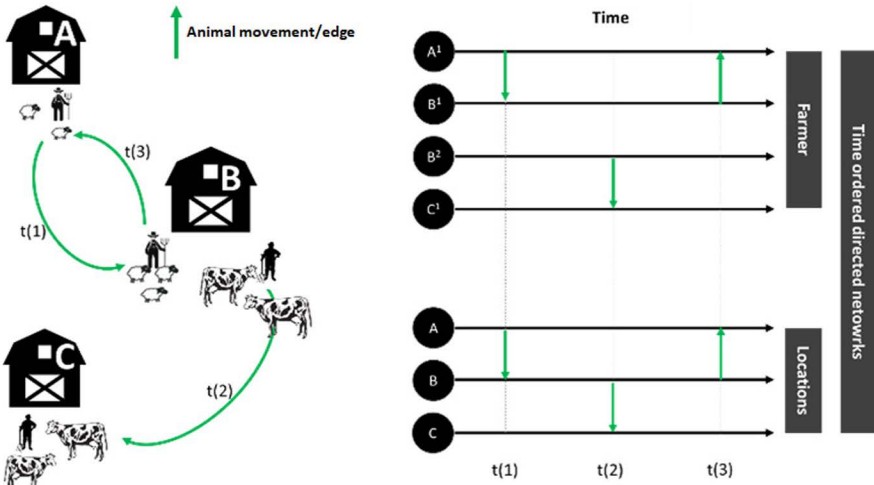

**Fig 1. Network fragmentation and aggregation in animal movement records.** The top right panel shows how multiple farmers of the same holding are represented as different nodes, leading to artificial fragmentation. The bottom right panel demonstrates the aggregation process, where each node represents the physical holding, encompassing multiple owners at a fiscal level.

the physical holding, i.e. the geographical location to where the animals transited. Thus, a node can represent multiple owners at a fiscal level (see Fig 1, bottom right).

The framework of time-ordered networks, essentially directed acyclic graphs [29], is useful for understanding flow dynamics [30], particularly in the context of diseases spread through direct contact. We assume diseases spread along paths driven by animal movements, so accounting for the direction and timing of these movements allows us to detect temporally valid paths, providing a likely estimation of disease spread, similar to the Susceptible-Infection network-based compartmental model (where the contact rate is one). Direction and the timing of animal movements are important for detecting disease spread because they highlight impossible paths, and or too-long time elapsed for short paths [30,31]. For example, in Fig 1, if farm A is infected at time zero, it can infect holding B, which can then infect holding C through subsequent animal movements. Since the probability of disease transmission decreases with increasing degrees of separation, we typically limit the search for plausible paths to a maximum degree of separation, or "level depth." Depending on the disease, the time elapsed between events might be too long to assume that nodes remain infectious and undetected, requiring a restriction of the time horizon to a specific number of days, known as "temporal depth." The choice of depth, in terms of both days and levels, affects the resulting subnetwork and depends on factors such as the disease's infectiousness, incubation period, and the timing of surveillance system controls. Furthermore, the removal of node(s) and/or edge(s) from a time-ordered directed network involves the change of the resulting paths. Therefore, each network modification requires a recalculation of the paths.

Contact tracing following a positive detection has been widely used in various fields, with different interpretations and applications depending on the context [30,32]. In T-Racing, FCT, BCT, and a combination of both (FCT-BCT) strategies have been implemented to support track and control the spread of diseases. These methods provide different perspectives and insights depending on the context and time of disease outbreaks.

**FCT method** focuses on identifying the pathways of pathogen spread starting from an index node, known as the seed, and an event date. By analyzing outgoing movements within a specified depth (in terms of days and levels) FCT highlights the nodes that the disease could have potentially reached. Here, levels represent the degrees of separation from the seed node, where each level corresponds to a different stage of contact tracing (e.g., direct contacts at Level 1, and contacts of contacts at Level 2). This approach supports targeted disease control more effectively than a general surveillance system, which, despite being timely conducted, can only prevent a portion of the potential disease spread (as illustrated in Fig 2). The control shown in Fig 2 can be considered a real-time monitoring and proactive control in response to a new outbreak detection. It is performed as soon as the new outbreak is notified to detect its outgoing movements in the recent past. The subnetwork found in a forward direction will help identify positive farms and prevent the spread of infection to other farms in the future.

**BCT method**, on the other hand, aims to trace the primary source of infection from a newly detected outbreak. This retrospective strategy consists of finding temporal valid pathways through incoming movements within specific time constraints and degrees of separation. Typically, BCT uses a broader time window compared to FCT, allowing for a comprehensive analysis of the infection's origins. Both methods rely on (require) updated data. Current Italian legislation requires the registration of animal movements in NDB within 7 days, but this does not always happen. More often, and especially for species regulated more recently, registration times are longer. This means that some of the real connections may be missing in NDB when performing FCT or BCT strategies, leading to undetected possible infectious nodes.

**Combination of strategies (FCT-BCT)** involves using both methods to enhance disease control efforts. For instance, after performing a BCT starting from multiple recent outbreaks

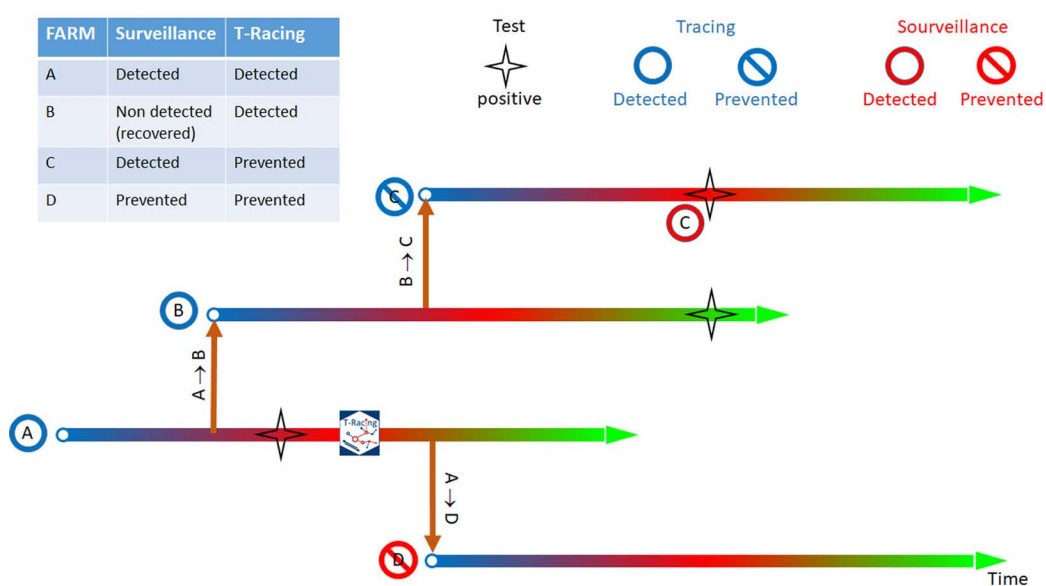

**Fig 2. Schematic illustration of the importance of tracing in preventing and detecting diffusion pathways versus the simple farm isolation when only positivity recorded in the context of the surveillance program is considered.**

within a large time window, the resulting subnetwork may include numerous nodes. If among these nodes there are outbreaks notified in the past, an FCT can be conducted on this newly identified BCT subnetwork. This intersection (BCT ∩ FCT) reveals the subnetwork connecting the outbreaks through temporally valid paths. Then, by working in the past, when data is more likely to be updated in NDB, this strategy identifies nodes that have been exposed to infected animals but have not yet tested positive. A comparison of animal movement dates with the last inspections on the farms can help promptly identify nodes for further inspection, aiding in containing the disease's spread. Ultimately, such kind of strategies could be performed using batches of animals (thus letting the paths highlight the spread accounting for transmission between animals sharing the same farms) or considering the individual animals moving forward or backward from a seed farm. These strategies can be applied either to batches of animals (highlighting the spread through shared farm contacts) or to individual animals moving between farms. The T-Racing application facilitates these analyses, enabling users to leverage epidemiological information and expert knowledge to transform vast amounts of data into clear, actionable guidelines for disease control.

## 2.3. T-Racing design

The T-Racing interface allows users to quickly query, filter, analyze, and visualize data through interactive graphs, maps, and tables for a seamless and intuitive experience. Fig 3 provides a conceptual map of the application, highlighting its primary dashboard sections and features.

### 2.3.1. Tools for extracting and viewing animal trade data.

- **Filtering data** based on multi-species selections, time frame, tracing direction mode (backward or forward methods), network geo-aggregation mode, and including/excluding animal trade data from/to foreign countries or slaughterhouses;

**Fig 3. T-Racing conceptual map.**

- **Multi-seed selection** for tracing, allowing users to either type or select one or more farm codes from the list of farms or select outbreak codes from the list of officially notified outbreaks within a specified time frame. Alternatively, the user can select all farms and/or outbreaks falling within a spatial buffer (neighbours distant by kilometres) around selected farms and outbreaks;

- **Tracking a single animal** by typing its identification code and viewing all resulting spatial-temporal paths within a specified time frame.

The system retrieves movement data based on these filters from the NDB and builds a time-ordered directed weighted network originated by the set seeds. These tools are useful for analyzing diseases that are transmitted through direct contact (i.e., brucellosis or tuberculosis) and diseases that are transmitted through spatial proximity (i.e., African swine fever or vector-borne diseases such as west Nile or Bluetongue, etc.), using spatial buffering and contact tracing methods.

### 2.3.2. Tools for customizing network visualization and optimization.

- **Customizing graph visualization** by setting from a range of layout options such as hierarchical, geographical, or nicely mode (default mode). Users can also select the shape and colour of nodes or the colour of edges to reflect chosen data attributes such as holding type or production type for nodes, and level, time, or species for edges. All possible attribute colours and shapes are predefined in a configuration file;

- **Removing** unnecessary **nodes or edges** from performed tracing;

- **Highlighting within the network all edges involving animals moved from/to the seed, or reducing the network to only those edges involving animals moved from/to the seed**, according to the tracing direction mode set by the user;

- **Reducing the network size** by decreasing network deepness (in days or levels).

For each network reduction method applied, a new time-ordered network will be recalculated to ensure and maintain the temporal coherence in the events.

### 2.3.3. Tools for epidemiological information integration and results printing.

- **Adding a layer** of sequenced pathogens for the animals, and/or of outbreaks for a specific disease **or reducing the resulting network** by keeping only temporally valid paths among involved outbreaks by a combination of FCT and BCT methods;

- **Download the network's tables** (based on selection) in Excel format;

- Storing the results in a **dynamic HTML report** for offline use.

By using these tools, the network graph can be customized and fine-tuned to provide a more meaningful representation of the data.

## 2.4. Data flow and system architecture

T-Racing has been developed using the Shiny library by the R statistical environment, and leaflet and visnetwork libraries were used for map and graph visualization respectively (Table 1 contains a list of all packages used). The application manages large amounts of data from different information systems and shares them efficiently and securely in real-time. Differently from the serverless JavaScript web apps, computationally heavy services are executed on a remote server, and the Plumber package allows the application to query data sources in real-time, through web API.

The system architecture facilitates the application scalability, thanks to shiny proxy (a free and open-source Docker-based application). The described stack and the involved application modules are displayed in Fig 4.

## 2.5. Demo

The T-Racing application is not publicly available as it handles sensitive data from Italian national information systems. It will be made available exclusively to local and national health officers at public and animal health agencies through the VETINFO portal. Therefore, a demo version has been developed solely for illustrative purposes to demonstrate its potential to the scientific community. This demo operates with a pre-configured dataset, which has been adjusted by adding noise and rounding coordinates to the second decimal place in the georeferencing of holdings, which may result in some holdings falling outside the Italian boundaries. The demo has been translated into English, while the original application is in Italian. The

**Table 1. List of the R-packages used in T-Racing development.**

| R-package | Use | Reference |
|---|---|---|
| shiny (v. 1.7.1) | UI/SERVER | Winston Chang, Joe Cheng, JJ Allaire, Carson Sievert, Barret Schloerke, Yihui Xie, Jeff Allen, Jonathan McPherson, Alan Dipert and Barbara Borges (2021) |
| shinyjs (v. 2.1.0) | UI/SERVER | Dean Attali (2020). shinyjs: Easily Improve the User Experience of Your Shiny Apps in Seconds |
| shinyWidgets (v. 0.6.4) | UI | Victor Perrier, Fanny Meyer and David Granjon (2021). shinyWidgets: Custom Inputs Widgets for Shiny |
| Shinydashboard (v. 0.7.2) | UI | Winston Chang and Barbara Borges Ribeiro (2018). shinydashboard: Create Dashboards with 'Shiny' |
| shinyBS (v. 0.61.1) | UI | Eric Bailey (2015). shinyBS: Twitter Bootstrap Components for Shiny |
| shinybusy (v. 0.3.0) | UI/SERVER | Fanny Meyer and Victor Perrier (2020). shinybusy: Busy Indicator for 'Shiny' Applications |
| DT (v. 0.22) | UI | Yihui Xie, Joe Cheng and Xianying Tan (2021). DT: A Wrapper of the JavaScript Library 'DataTables' |
| rintrojs (v. 0.3.0) | UI/SERVER | Carl Ganz (2016). rintrojs: A Wrapper for the Intro.js Library. Journal of Open Source Software, 1(6), October 2016 |
| shinycssloaders (v. 1.0.0) | UI/SERVER | Andras Sali and Dean Attali (2020). shinycssloaders: Add Loading Animations to a 'shiny' Output While It's Recalculating |
| htmltools (v. 0.5.2) | UI | Joe Cheng, Carson Sievert, Barret Schloerke, Winston Chang, Yihui Xie and Jeff Allen (2021). htmltools: Tools for HTML |
| leaflet (v. 2.1.1) | UI | Joe Cheng, Bhaskar Karambelkar and Yihui Xie (2022). leaflet: Create Interactive Web Maps with the JavaScript 'Leaflet' Library |
| rgdal (v. 1.5.31) | SERVER | Roger Bivand, Tim Keitt and Barry Rowlingson (2021). rgdal: Bindings for the 'Geospatial' Data Abstraction Library |
| sp (v. 1.4.7) | SERVER | Pebesma, E.J., R.S. Bivand, 2005. Classes and methods for spatial data in R. R News 5 (2) |
| bezier (v. 1.1.2) | SERVER | Aaron Olsen (2018). bezier: Toolkit for Bezier Curves and Splines |
| geosphere (v. 1.5.14) | SERVER | Robert J. Hijmans (2021). geosphere: Spherical Trigonometry |
| visNetwork (v. 2.1.0) | UI | Almende B.V., Benoit Thieurmel and Titouan Robert (2019). visNetwork: Network Visualization using 'vis.js' Library |
| igraph (v. 1.3.1) | SERVER | Csardi G, Nepusz T: The igraph software package for complex network research, InterJournal, Complex Systems 1695. 2006 |
| ggplot2 (v. 3.3.6) | SERVER | H. Wickham. ggplot2: Elegant Graphics for Data Analysis. Springer-Verlag New York, 2016 |
| RColorBrewer (v. 1.1.3) | SERVER | Erich Neuwirth (2014). RColorBrewer: ColorBrewer Palettes |
| xlsx (v. 0.6.5) | SERVER | Adrian Dragulescu and Cole Arendt (2020). xlsx: Read, Write, Format Excel 2007 and Excel 97/2000/XP/2003 Files |
| dplyr (v. 1.0.9) | SERVER | Hadley Wickham, Romain François, Lionel Henry and Kirill Müller (2021). dplyr: A Grammar of Data Manipulation |
| jsonlite (v. 1.8.0) | SERVER | Jeroen Ooms (2014). The jsonlite Package: A Practical and Consistent Mapping Between JSON Data and R Objects. arXiv:1403.2805 |
| curl (v. 4.3.2) | SERVER | Jeroen Ooms, Hadley Wickham, RStudio |
| rJava (v. 1.0.6) | UI | |

original development and publication environment, as well as the web services, have been duplicated and adapted for the demo. The underlying dataset is contained in a single Oracle database. The demo version is intended to allow all readers of this article to explore and replicate the results presented in our study, providing a comprehensive preview of its features as listed in Table 2.

## Architecture

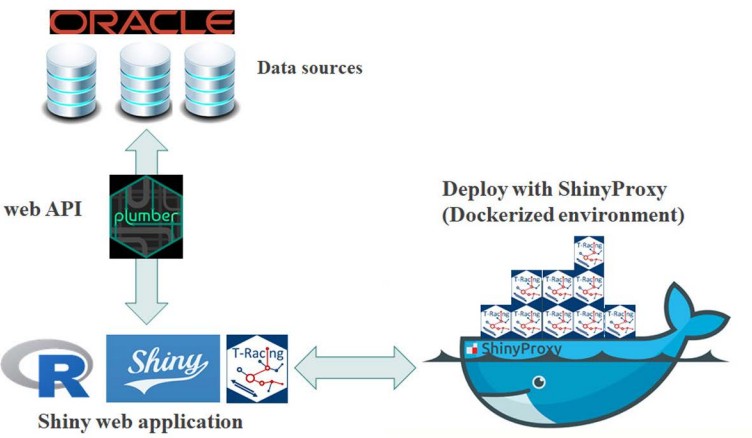

**Fig 4. T-Racing architecture.**

**Table 2. List of data available in T-racing demo.**

| Data | From | To | Type/feature |
|---|---|---|---|
| Movements | 01/12/2017 | 30/06/2023 | batch and individual for bovine and buffalo |
| Outbreaks | 01/12/2017 | 30/06/2023 | *Brucellosis, Tuberculosis* |
| ST | 01/01/2015 | 01/12/2018 | *B. abortus* and *B. melitensis* species |

### 2.6. Brucellosis and tuberculosis: rationale for case studies and Italian regulations

To demonstrate the real-world utility of the application the following case studies were considered:

**Brucellosis in the Marche region (central Italy).** On December 9, 2019, a brucellosis outbreak was confirmed in central Italy, specifically in the Marche region. The T-Racing application was utilized to identify the potential source of the infection. Only bovine species were considered, while slaughterhouses and foreign countries were excluded from the analysis. The known outbreak code from a list of proposed brucellosis outbreaks was selected as the seed for a trace backwards analysis encompassing all incoming movements to the identified outbreak over the past two years at 4 depth levels. The resulting network was refined using the "outbreaks connection" tool, and genetic data was added to support the conclusions drawn from the final network.

**Tuberculosis outbreaks affected Elba Island and the nearby Tuscany coast** in June 2023. A forward tracing from March to December 2023 to identify the likely spread trajectory leading up to the outbreak on Elba Island was used. Only bovine species were considered, slaughterhouses were included, and foreign countries were excluded from the analysis. Additionally, we utilized a "Seed animal's path highlighting" tool to emphasize the flow of the infection about the animals coming from the seed.

**A tuberculosis outbreak occurred in the Cuneo province in December 2022.** A trace backward analysis over a 13-month temporal window with a level depth of 4 was used to determine the most likely source of infection. Network reduction was applied using the dedicated tools for highlighting paths among involved outbreaks ("outbreaks connection" tool).

We used T-Racing to investigate brucellosis and tuberculosis outbreaks in non–endemic areas of Italy, where an occurrence of an outbreak should trigger a rigorous investigation by the official AH service to prevent the disease from spreading, demonstrating the speed and effectiveness of the investigation carried out with the proposed web app. Brucellosis caused by *Brucella abortus* is an important zoonoses worldwide, and tuberculosis caused by *Mycobacterium bovis* constitute a serious hazard to PH in developing countries and a great economic burden to livestock production in Europe. These diseases were chosen due to their PH and economic importance and their chronic development, allowing for the use of FCT and BCT methods with a broad time-elapse. Prevention of these zoonoses in humans depends on the control of the disease in animals. Livestock movement data represent a valuable source of information to understand the pattern of contacts between holdings, which may determine the spread of such diseases. The first national plans for the control of bovine tuberculosis and brucellosis were approved with the Ministerial Decree of 11 and 12 March 1965, respectively. Through the Ministerial Decree of 27 August 1994 n. 651 entered into force in Italy the national eradication plan for bovine brucellosis, which establishes the measures to be applied to cattle farms to obtain the eradication of the disease for the protection of PH and protection of officially free herds. The regulation determines for control the use of official serological tests twice a year at an interval of not less than three and not more than six months. All animals over the age of twelve months in all cattle farms have to be tested, even if are located in the wild. Similar procedures have been established (test all animals in all heads at once, with 3 to 6 months intervals) with Ministerial Decree 15 December 1995, n. 592 – Regulation concerning the national plan for the eradication of tuberculosis in cattle and buffalo herds – by testing with the tuberculin intradermal test.

## 3. Results

T-Racing Demo, which uses a limited and anonymized dataset, is available at this link https://demo.izs.it/app_direct/tracing/. The basemaps throughout all the following figures were created using ArcGIS software by Esri and integrated into the shiny web application using the leaflet.providers R library, sourced from leaflet-providers.js (https://github.com/leaflet-extras/leaflet-providers). ArcGIS and ArcMap are the intellectual property of Esri and are used herein under license. Copyright Esri. All rights reserved. For more information about Esri software, please visit www.esri.com.

Fig 5 shows the T-Racing interface, which displays temporally valid paths originating from a brucellosis outbreak in Sicily, an Italian southern island, using the trace forward method. The investigation revealed this outbreak, detected in July 2020, was likely the source of two subsequent outbreaks in Pisa, Tuscany, northern Italy. Animals moved from the seed in March reaching, via a transit station, two farms in Pisa in April and May, but the outbreaks were only detected by surveillance in December 2020. The simultaneous visualization of the queryable and interactive graph, map, and tables, combined with the iterative use of tools for network calibration and targeted result personalization, offers a more comprehensive and user-friendly result viewing.

### 3.1. T-Racing - case studies

**3.1.1. Tracing backward from a notified outbreak of brucellosis in the Marche region of central Italy, in December 2019, to identify its potential sources of infection.** The search for the subnetwork resulted in 1.825 movements involving 386 nodes. The subnetwork covers a substantial area of central Italy, including holdings that reach the seed even from the islands of Sicily and Sardinia. The theming of nodes based on type reveals the involvement of different holding types. In addition to breeding farms, there are pastures (structures that play a significant role in the spread of brucellosis) and

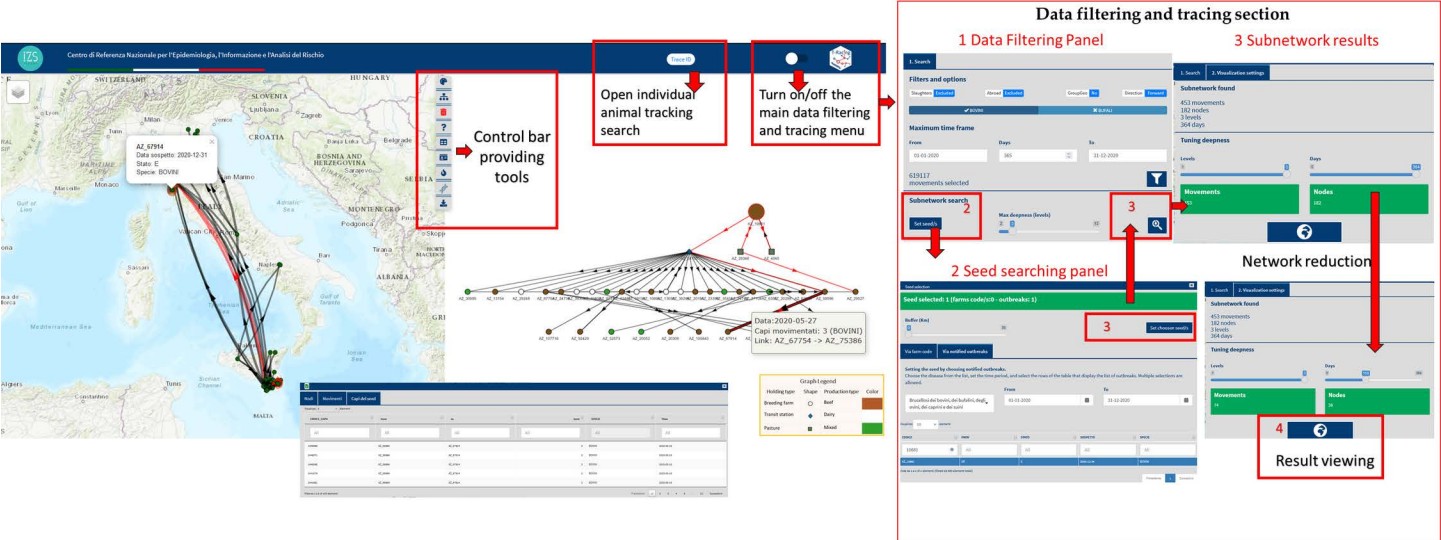

**Fig 5. The T-Racing interface (on the left) shows the subnetwork resulting from the trace forward analysis on a map, graph, and tables and the steps (1–4) followed to get the results in the interface are shown on the right.** This analysis was performed using a brucellosis outbreak in July 2020 in the Sicily Region as the seed, extracting all bovine movements in 2020 up to the third level of depth (three degrees of separation). A time frame-based network reduction/optimization method was applied for the result. The "hierarchical" graph layout shows all nodes and edges in the subnetwork, highlighting probable disease pathways and nodes reached during the study period. Node shape and colour represent farm type and production activity (see legend). The larger, red-bordered icon represents the seed, with each node labelled by farm code. Arrows indicate animal movements between two nodes, with the width proportional to the number of animals moved. Red-coloured arrows highlight paths followed by animals moving from the seed (using the "Seed animals path highlighting" tool). Fire icons on the map represent brucellosis outbreaks detected during the selected period and included in the subnetwork (using the "Outbreaks connections" tool). Tooltips and tables provide detailed information, facilitating navigation between the map, graph, and tables. The Basemap throughout this figure were created using ArcGIS software by Esri. ArcGIS and ArcMap are the intellectual property of Esri and are used herein under license. Copyright Esri. All rights reserved. For more information about Esri software, please visit www.esri.com.

a hub node, a genetic collection centre located in Perugia province, Umbria region, in central Italy, that connects 132 nodes (Fig 6). To make the network more manageable and relevant for the conducted tracing, a subnetwork was created by setting 3 levels and reducing the depth to 399 days, resulting in 198 movements and 38 nodes. To identify plausible paths of disease spread, the application allows the utilization of notified outbreaks within a customizable time frame for the specific disease. These outbreaks are visualized on the map, enabling the tracing to be narrowed down to the connections between outbreaks included in the tracing. This helps identify a focused subset of connections to investigate. The subnetwork includes multiple brucellosis outbreaks, primarily located in the southern and central regions. The reduced network, created by keeping only temporally valid paths among involved outbreaks within the network, consists of 94 movements involving 11 nodes. At this point, we can assess the presence of genetic information as further confirmation of the tracing's plausibility. To do so, we utilize the tool by adding the available sequence type for *Brucella abortus*. The seed is an outbreak for which sequence type "2" is available, and the identified origin zones are also affected by clusters of the same sequence type. An interesting aspect is to determine how many of these links involve the same animals that reached the seed. The dedicated tool allows for theming the links based on this characteristic by searching for individually identified animals. The visualization highlights on the subnetwork the paths involving the same animals that reach the seed. These highlights paths consist of only 21 movements involving 5 nodes (including the seed and one pasture, which is the presumed source of infection). Further investigation of individually identified and traced animals revealed that

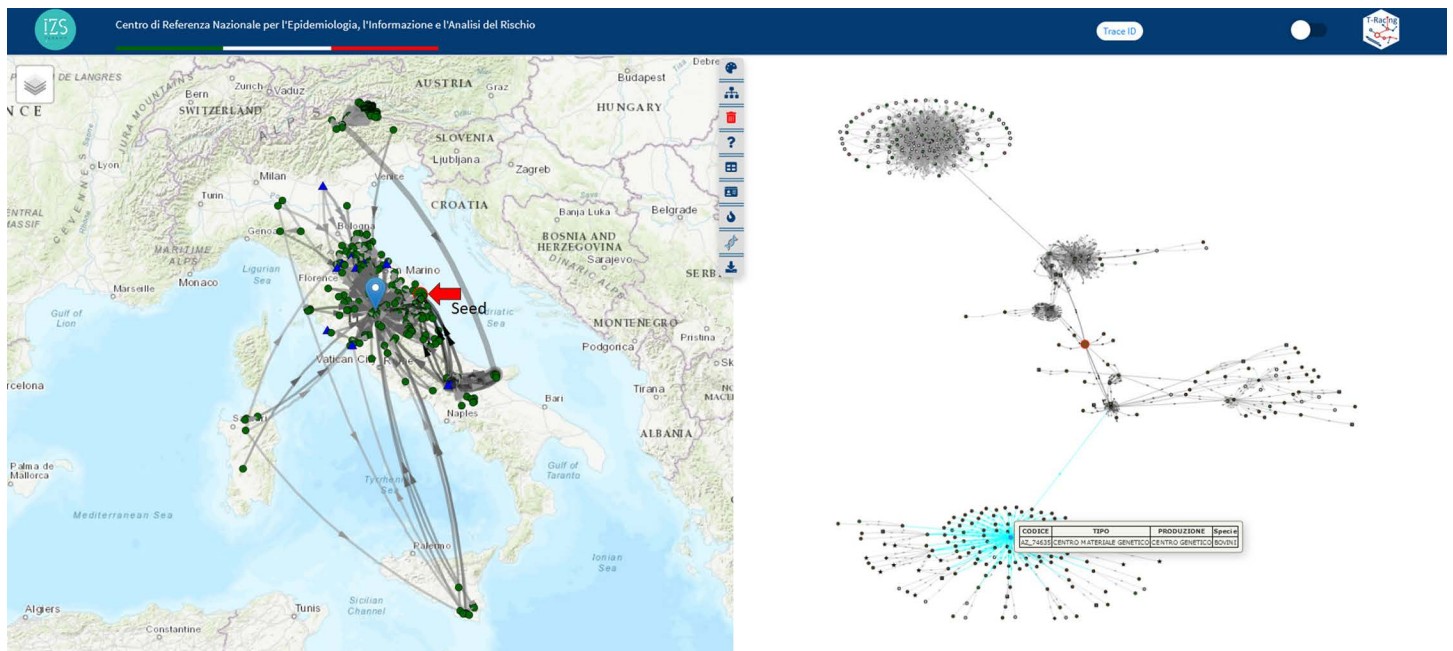

**Fig 6. Trace backward method.** Sequential investigation steps: 1) Data Filter, 2) Set seed by outbreak code search, and 3) Network view. The trace-back focusing on bovine species, excluding slaughterhouses and foreign countries, was conducted to trace the outbreak occurring in December in the Marche region, in Italy. 1,825 movements involving 386 nodes were identified within a broad time spanning the past two years. The seed is contoured in red. The network covers a substantial area of central Italy, including farms extending to the islands (Sicily and Sardinia). Node theming based on structure type highlights diverse holding types, including pastures and a genetic collection centre in Perugia, highlighted in light blue on the graph (simultaneously, its marker appears on the map). The Basemap throughout this figure were created using ArcGIS software by Esri. ArcGIS and ArcMap are the intellectual property of Esri and are used herein under license. Copyright Esri. All rights reserved. For more information about Esri software, please visit www.esri.com.

an infected animal, identified as sequence type "2" in the seed, originated from a pasture, which was subsequently identified as the plausible source of the sought-after infection (Fig 7).

**3.1.2. Tracing the flow of tuberculosis outbreaks: Elba Island and Cuneo Province. - Four BTB outbreaks were notified in June 2023 in a small, non-endemic area, in Italy, affecting Elba Island and the nearby Tuscany coast.** Using T-Racing allowed for the detection of the real flow of the disease spread. The resulting network highlighted an animal movement of nine cattle in April 2023, which reached farms on the nearby coast. This established the real source of the infection and the most likely flow of the spread (Fig 8). We discovered that the animals reaching the outbreaks were the same ones that had left Elba Island and one of them (at the time of writing this article) had not yet been slaughtered. **-In December 2022, an outbreak of BTB occurred in the Cuneo province of the Piedmont region in northern Italy.** The resulting network comprised 306 farms and 664 connections, predominantly located in the Sicily region. For a more focused investigation, we considered only temporally valid paths connecting BTB outbreaks included in the network resulting in a subnetwork of 50 connections involving 9 breeding farms in the Sicily region, four of which were BTB outbreaks, and two pastures. The investigation suggested that four outbreaks occurring in the previous months are the most likely source of infection. Additionally, the 119 animals reaching the Cuneo province in June and July originated from a fattening farm where they were likely infected by mixing with animals coming from the four outbreaks. The results indicate the need for targeted control measures for the other five farms involved in the revealed diffusion paths (Fig 9).

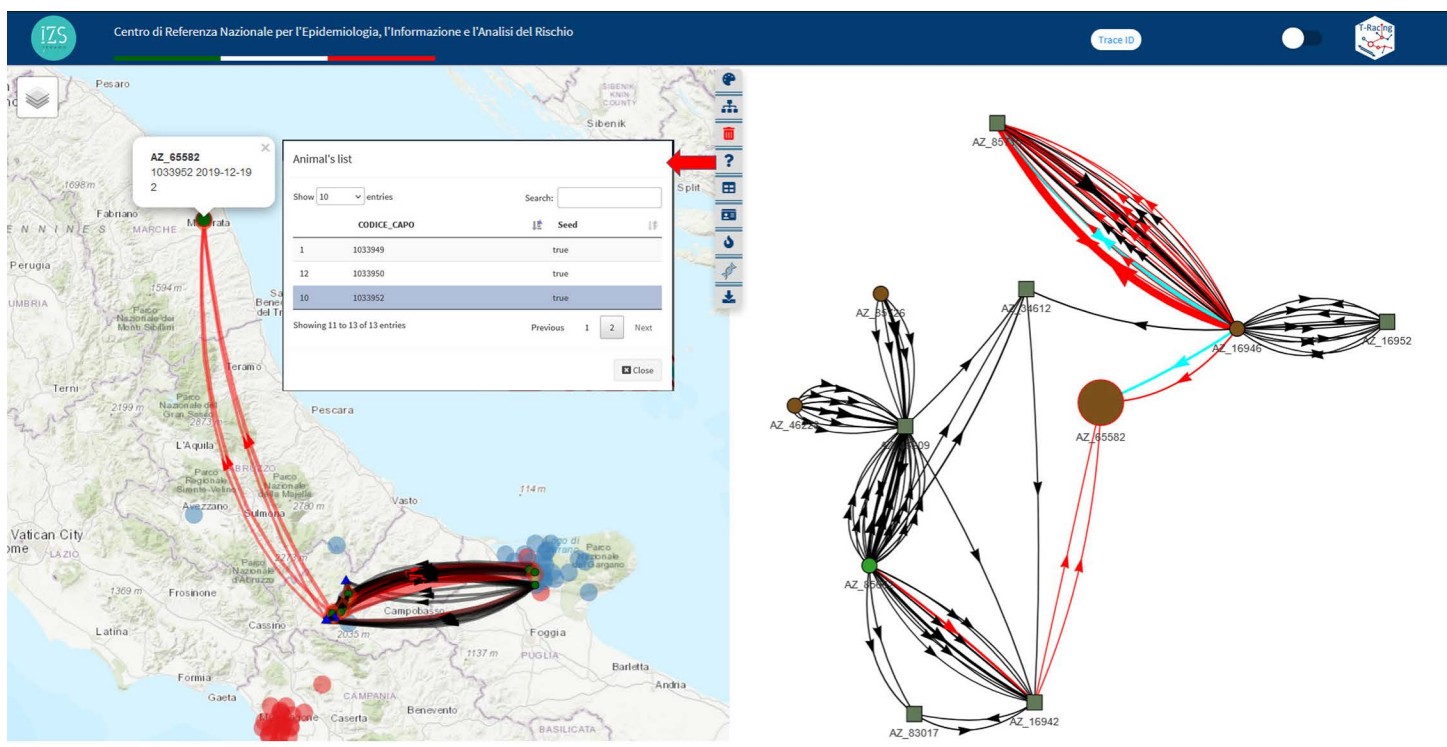

**Fig 7. The subnetwork result after the following four sequential steps: 1) Network reduction by setting the network level to 3 and days depth to 399 (the new subnetwork size is 198 links and 38 nodes), 2) Further reducing the subnetwork by keeping only temporally valid paths among involved outbreaks (resulting in a subnetwork size of 94 links and 11 nodes; the four detected brucellosis' outbreaks involved are shown on the map with red fire icons), 3) Adding the sequence type for Brucella abortus layer to the map (type 2 in blue circles) and displaying a popup on the seed showing the identification code of the animal, sampled date, and detected Brucella abortus sequence type, and 4) Highlighting on the graph are all links involving the same animals that reached the seed (red links).** Further investigation on "individually identified animals," found by querying the highlighted links (in light blue on the graph), revealed that an infected animal originating from the pasture reached the seed. This animal is listed in the Animal list produced by the application, matching the animal reported in the map popup. The Basemap throughout this figure were created using ArcGIS software by Esri. ArcGIS and ArcMap are the intellectual property of Esri and are used herein under license. Copyright Esri. All rights reserved. For more information about Esri software, please visit www.esri.com.

## 4. Discussion

The T-Racing application was created by a team of experts including epidemiologists, veterinarians, mathematicians, and computer scientists. This collaborative effort spanned all stages of development, from gathering data to creating tools and building a user-friendly interface. The team has extensive experience in epidemiological tracing and dynamic network analysis, as evidenced by similar web applications they have developed since 2011 [24,33–36]. T-Racing represents an improved version of previous applications, offering faster speeds, additional features, and integration of various epidemiological datasets. Currently, it is available on the internal network of IZS-Teramo and used by our veterinarians and epidemiologists. It is pending technical adjustments and official approval from the Italian Ministry of Health to be definitively integrated into the VETINFO portal, where T-Racing will inherit both the users and the robust GDPR techniques already applied in the portal.

T-Racing offers advantages in terms of timely identification of outbreaks and systematic contact tracing, which can be effective in stopping the transmission chain of contagious diseases. This is particularly effective when strong collaborations exists between epidemiological teams, laboratory surveillance, and decision-making officials, improving outbreak management, in a multidisciplinary context [37].

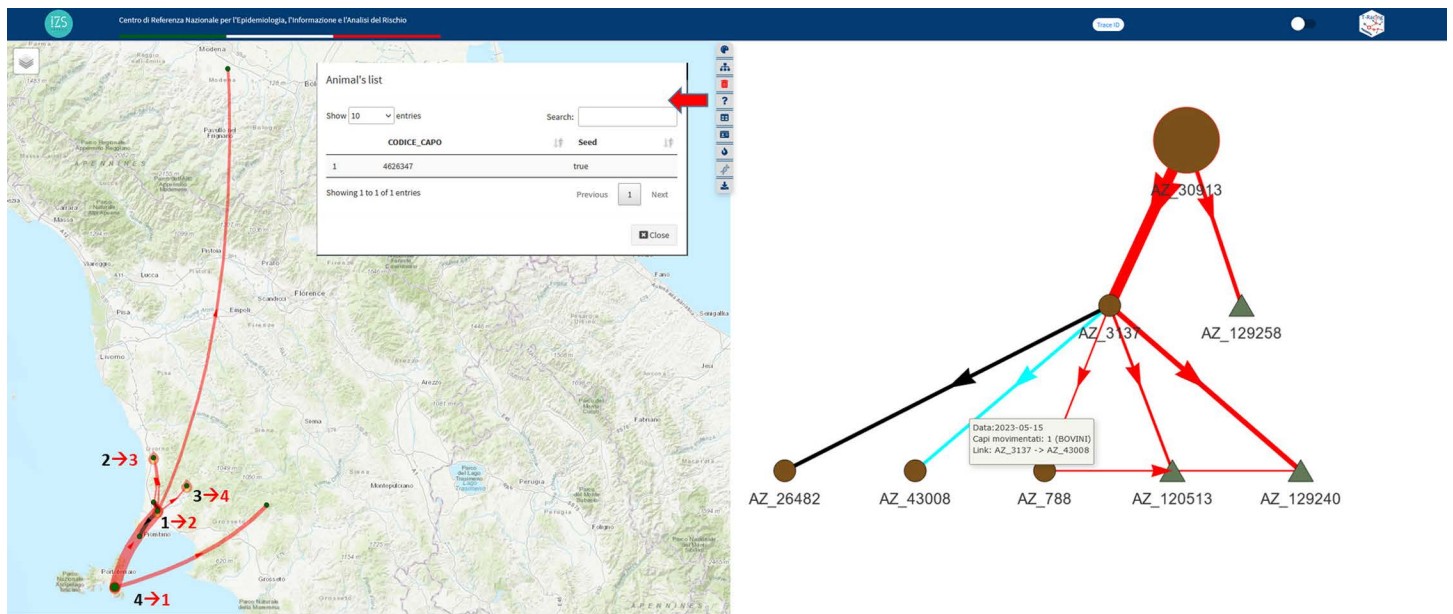

**Fig 8. Trace forward method.** The map and graph show outbreaks that occurred on Elba island and the nearby Tuscany coast (black numbers '1' to '4', represent the notification timing, with 1 the earliest and '4' the most recent notified one). The network of animal movements involving the outbreaks is displayed with red links indicating the presence of at least one animal originating from the seed, and black link representing the movement of animals not originating from the seed. The notified outbreaks' timing contrasts with the real flow of disease spread revealed by T-Racing analysis (red numbers '1-4'). The triangles on the graph are slaughterhouses. Further investigation on "individually identified animals," found by querying the highlighted link (in light blue on the graph), revealed that one animal originating from the seed had not yet been slaughtered. This animal is listed in the Animal's list produced by the application. The Basemap throughout this figure were created using ArcGIS software by Esri. ArcGIS and ArcMap are the intellectual property of Esri and are used herein under license. Copyright Esri. All rights reserved. For more information about Esri software, please visit www.esri.com.

The capability of T-Racing to handle zoonotic diseases such as brucellosis, tuberculosis, and others under AH and PH surveillance in Italy highlights its crucial role in preventing and controlling the spread of infections across species. This also supports broader PH initiatives, as effective control of animal outbreaks, facilitates coordinated response strategies. In Italy, both AH and PH services operate under the Ministry of Health, fostering close collaboration between these sectors. This structural collaboration between PH and AH authorities, which is unique in our national system, supports unified efforts in zoonotic disease management, environmental disaster responses, and food safety, enabling a smoother adoption of the One Health approach.

T-Racing has the potential to significantly enhance animal disease control measures, as it can rapidly organize crucial contact information from extensive datasets and identify connections in a multi-outbreak investigation. It introduces new methods and tools currently not available in other systems or tools [2,24,33–36]. These include multi-seed/species selection, seed selection using a spatial buffer around cases or outbreaks, and the combination of strategies (FCT-BCT) configuring itself as an innovative application in its field. Thus, the application efficiently supports epidemiological investigations, as it was shown in the examples for brucellosis and tuberculosis. In these examples, it is possible to see how deep into the transmission chain the T-Racing allows us to explore. The initial tracing example identified a likely path of disease spread and the presumed origin, starting from a relatively large and diverse network (1.825 x 386) in both spatial and temporal dimensions. By using interactive and iterative tools, along with additional information such as disease outbreaks, genetic data, and movements of individually identified animals, it was possible to narrow down the focus to 21 movements and 5 nodes in just a few steps, saving a significant amount of time. Trying

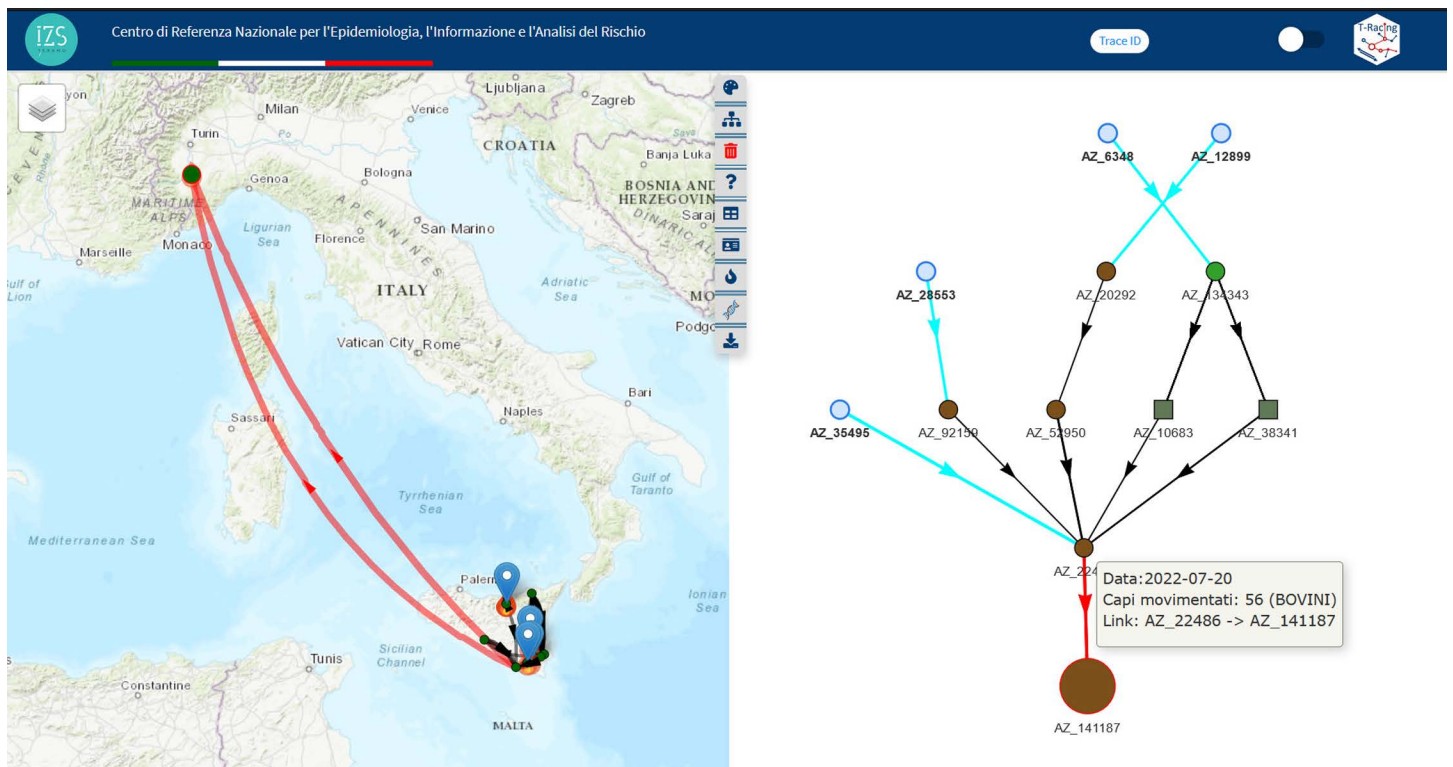

**Fig 9. Trace backward method.** The network resulting from the BTB outbreak occurred in Cuneo province, in the Tuscany region in December 2022 involving 306 farms and 664 links, mostly located in the Sicily region. The map and graph show the reduced network by considering only temporally valid paths connecting involved outbreaks consisting of 50 connections and 12 holdings which 9 breeding farms in the Sicily region (circles on the graph), four of which are BTB outbreaks (fire icons and marked on the map and highlighted in light blue on the graph), and two pastures (squares on the graph). The links involving animals reaching the Cuneo outbreak are in red colour.

to achieve the same result by simply extracting the movement network through a query of the NDB would have been extremely difficult, like finding a needle in a haystack. The second example clarified the actual timing of the tuberculosis spread. Importantly, the animals from the seed had not been slaughtered at the time of writing this article, and in the last example, results indicate a need for targeted control measures for the other five farms involved in the revealed diffusion paths.

Considering chronic diseases, such as brucellosis and tuberculosis, clarifying the disease's spread patterns allows for focus measures of control, improving targeting financial and human resources, as well as appropriate risk assessment and decision-making. Moreover, the almost real-time data used, the possibility of analyzing many datasets at once and the user-friendly environment are also features that could greatly advance responses to emergencies with acute and highly contagious diseases when movement data could be available in real-time.

The integration of molecular epidemiological data into the analysis complements the validation of identified disease (brucellosis) transmission paths, exemplifying the approach developed by an interdisciplinary team to advance disease control technology. Currently, the T-Racing application does not integrate genomic data from a variety of sources, such as human, environmental, and vector samples. However, incorporating such data in the future would be beneficial for enhancing the One Health approach and improving overall disease management. T-Racing empowers field epidemiologists, AH workers, researchers, risk assessors, and decision-makers at AH and PH agencies to leverage animal surveillance data

for more comprehensive, integrated disease control strategies [3,38–41], enabling informed decisions and proactive measures to mitigate the impact of zoonotic diseases. This tool's data-driven nature underscores the importance of accurate data management, as reliable data is crucial for effective and trustworthy epidemiological investigations. As T-Racing evolves, we aim to connect it with existing and future public health information systems through web services.

Although T-Racing is currently operational, it does have some limitations that are important to consider. The application's specific structure, tailored to the Italian context, limits its broader usability. At present, the genetic data is restricted to the Brucella genus, but additional pathogen data will be incorporated as soon as it becomes available in GenPat. Data regarding testing from surveillance systems on farms will be included, and functions will be created to factor in the timing of testing when selecting the most probable disease spread paths. Also, detailed information about single animals, like demographics and parturition dates for calves, along with human cases will be considered. An evaluation is also underway for the development of scheduled algorithms that will operate directly on the database to identify critical networks to be automatically provided to T-Racing's users. This will enable more efficient, rapid, and straightforward validation and initiation of investigations. However, the authors and team of experts will continue to develop future versions to improve the application by adding new datasets and functionalities. Concurrently, work is ongoing for a more general application that allows users to upload their own data, and its source code will be freely distributed on GitHub.

## 5. Conclusions

The T-Racing web app represents a modern and forward-thinking approach to support epidemiological investigations in animal disease outbreaks in Italy. It provides a unified environment to manage and analyze the expanding pool of data sources iteratively by integrating advanced technology, comprehensive data management, and cutting-edge analytics. T-Racing has the capability to draw meaningful conclusions based on a data-driven, user-guided approach while leveraging its expertise. It facilitates evidence-based decision-making and serves as a leading tool in disease control, strengthening our capacity to address emerging challenges and safeguard PH. In the future, expanding the integration of PH data will further enhance its potential in the One Health framework.

## Author contributions

**Conceptualization:** Luca Candeloro, Lara Savini, Annamaria Conte, Paolo Calistri, Fabrizio De Massis.

**Data curation:** Diana Palma.

**Funding acquisition:** Paolo Calistri.

**Methodology:** Luca Candeloro, Lara Savini.

**Project administration:** Paolo Calistri.

**Software:** Luca Candeloro, Samuel Perticara, Anna Cecilia Trolesi Reis Borges Costa, Karina Diniz Baumgarten.

**Validation:** Paolo Calistri, Fabrizio De Massis.

**Writing – original draft:** Luca Candeloro, Lara Savini, Fabrizio De Massis.

**Writing – review & editing:** Luca Candeloro, Lara Savini, Annamaria Conte, Paolo Calistri, Diana Palma, Samuel Perticara, Anna Cecilia Trolesi Reis Borges Costa, Alessio Di Lorenzo, Susanna Tora, Sara Serrani, Michela Toro, Karina Diniz Baumgarten, Fabrizio De Massis.

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
