## [Decision Letter · Decision Letter 0]

8 Mar 2024

PONE-D-24-01354T-Racing: a modern tool for supporting epidemiological investigation in animal disease outbreaksPLOS ONE

Dear Dr. Candeloro,

Thank you for submitting your manuscript to PLOS ONE. After careful consideration, we feel that it has merit but does not fully meet PLOS ONE’s publication criteria as it currently stands. Therefore, we invite you to submit a revised version of the manuscript that addresses the points raised during the review process. All the reviewers highlight major concerns with the manuscript. I have set a long revision time to allow authors address on the raised points,

It must be noted that a paper describing a tool cannot be only based on its description/simulation, but must provide evidence of a working instance.

I invite the authors to thoroughly revise the work, with major focus on

1) providing a working instance of the tool (online or as a stand-alone application to be downloaded),

2) restructuring the organization and narrative of the manuscript.

We look forward to receiving your revised manuscript.

Kind regards,

Anna Bernasconi, PhD  

Academic Editor

PLOS ONE

“This research was funded by Italian Ministry of Health (MSRCTE0219).”

5. Please update your submission to use the PLOS LaTeX template. The template and more information on our requirements for LaTeX submissions can be found at http://journals.plos.org/plosone/s/latex.

6. We note that you have indicated that there are restrictions to data sharing for this study. PLOS only allows data to be available upon request if there are legal or ethical restrictions on sharing data publicly. For more information on unacceptable data access restrictions, please see http://journals.plos.org/plosone/s/data-availability#loc-unacceptable-data-access-restrictions.

7. One of the noted authors is a group or consortium [Italian Ministry of Health]. In addition to naming the author group, please list the individual authors and affiliations within this group in the acknowledgments section of your manuscript. Please also indicate clearly a lead author for this group along with a contact email address.

8. Please amend either the abstract on the online submission form (via Edit Submission) or the abstract in the manuscript so that they are identical.

9. We note that Figure 1 in your submission contain copyrighted images. All PLOS content is published under the Creative Commons Attribution License (CC BY 4.0), which means that the manuscript, images, and Supporting Information files will be freely available online, and any third party is permitted to access, download, copy, distribute, and use these materials in any way, even commercially, with proper attribution. For more information, see our copyright guidelines: http://journals.plos.org/plosone/s/licenses-and-copyright.

10. We note that Figures 2, 3, 4, 5, 6 , and 7 in your submission contain [map/satellite] images which may be copyrighted. All PLOS content is published under the Creative Commons Attribution License (CC BY 4.0), which means that the manuscript, images, and Supporting Information files will be freely available online, and any third party is permitted to access, download, copy, distribute, and use these materials in any way, even commercially, with proper attribution. For these reasons, we cannot publish previously copyrighted maps or satellite images created using proprietary data, such as Google software (Google Maps, Street View, and Earth). For more information, see our copyright guidelines: http://journals.plos.org/plosone/s/licenses-and-copyright.

1. You may seek permission from the original copyright holder of Figures 2, 3, 4, 5, 6 , and 7 to publish the content specifically under the CC BY 4.0 license. 

Additional Editor Comments:

All the reviewers highlight major concerns with the manuscript. I have set a long revision time to allow authors address on the raised points,

It must be noted that a paper describing a tool cannot be only based on its description/simulation, but must provide evidence of a working instance.

I invite the authors to thoroughly revise the work, with major focus on

1) providing a working instance of the tool (online or as a stand-alone application to be downloaded),

2) restructuring the organization and narrative of the manuscript.

Reviewers' comments:

Reviewer's Responses to Questions

**Comments to the Author**

1. Is the manuscript technically sound, and do the data support the conclusions?

Reviewer #1: Yes

Reviewer #2: Partly

Reviewer #3: Partly

Reviewer #4: Partly

2. Has the statistical analysis been performed appropriately and rigorously? 

Reviewer #1: N/A

Reviewer #2: I Don't Know

Reviewer #3: N/A

Reviewer #4: I Don't Know

3. Have the authors made all data underlying the findings in their manuscript fully available?

Reviewer #1: Yes

Reviewer #2: No

Reviewer #3: No

Reviewer #4: No

4. Is the manuscript presented in an intelligible fashion and written in standard English?

Reviewer #1: Yes

Reviewer #2: Yes

Reviewer #3: No

Reviewer #4: No

5. Review Comments to the Author

Reviewer #1: Dear Authors,

I enjoyed having the opportunity to review this paper. In summary, this study describes a tool the authors have built that draws on multiple sources of data allowing the user to explore animal trade networks. Coupled with surveillance data, this acts as a tracing system to expedite the control of disease outbreaks. Real-world examples are cited for zoonotic diseases; Brucella abortus and Mycobacterium bovis, to give an example of the One Health utility of this tool. I have a number of comments, corrections and observations. The tool you have created and described has similar counterparts in other countries, often aggregated into an overall Emergency Disease Management System – of which I’m am aware of a few. These are usually commissioned by Agriculture ministries and designed and implemented by IT professionals. As such, their design and function rarely sees light in the scientific public domain. For this reason, I consider this piece of work useful to showcase. As such I am recommending this paper be accepted with revisions. I have chosen minor revisions because the content of what you are presenting is sound however, there is quite a bit of editorial work required to improve the general flow of the paper.

The presentation of this paper is generally good however the flow of writing is interrupted by the breaking up of continuous sentences into paragraphs unnecessarily. The manuscript would be greatly improved by collapsing these paragraphs appropriately. There is also a degree of repetition which could be pared back.

The insertion of the case studies is essential for describing the tool but the subject matters, BTB and Brucellosis, aren’t be properly described in terms of their aetiology, natural history, etc.; nor can they be. I’m not sure how to address this. Possibly a single paragraph on each disease as part of the case study introduction?

It is inferred in the text, but I think there should be more a more explicit description of how the user’s expert elicitation is required to drive the tool. The tools helps guide the user rather than offering an instant solution.

I assume the tool is a work-in-progress? This should also be addressed. Are their future plans for improvements/enhancements? Can the tool be leveraged to provide researchers with data on animal contacts of disease modelling, etc? These would make useful discussion points.

Brucella spread usually has a strong temporal component. The lead up to the calving season is the highest risk time for spread of infection. Could a temporal window be applied to tracings to add a risk weighting to cohorts of animals residing together at these times? This could help narrow down the number of nodes.

The Brucella case study cited genetic sequencing as an additional layer to confirming tracings. Has this been done with BTB in this study?

The stated sensitivity of the skin test is at an animal level. See text in manuscript comments.

The figures are quite small and pixelated. Perhaps this is just in the review version (pdf)?

The following comments relate to editorial changes within the manuscript:

Line 10: ‘and lets’ suggest change to ‘helping’

Line 11: ‘to safeguard’ suggest change to 'of safeguarding’

Line 13: ‘multiple simultaneously and’ change to ‘multiple sources simultaneously at’

Line 14: delete ‘sources’

Line 14: ‘has been’ change to ‘is’

Line 18: Delete ‘some’

Line 27: ‘reduce’ to ‘reduces’

Line 8, 33, 311,312, 320, etc. One Health or One-health or One-Health?! Stick with one!

Line 36 ‘data is’ change to ‘data are’

Lines 48. 52. 57. 60. 63. etc. New paragraphs created unnecessarily. Remove paragraph breaks and associated indentation where not required.

Line 55. ‘require to rapidly’ change to ‘require to be able to rapidly’

Line 63. ‘of disease spreading’ suggest change to ‘within the spread of disease’

Line 69. ‘who’ change to ‘that’

Line 70. ‘virus’ change to ‘disease’

Line 72. ‘This could be an invaluable tool that allows’ change to ‘This would be an invaluable tool allowing’

Line 76. ‘interacting’ change to ‘interactive’

Line 80. ‘accounting for the’ change to ‘utilizing’

Line 82. Delete ‘Along with’ and don’t start a new paragraph.

Line 102. ‘(in Table 1 are listed all of the packages used)’ change to ‘(Table 1 contains a list of all packages used)

Line 102 – 103 ‘The application lets manage large amounts of data from different information systems and share them efficiently and securely in real-time.’ Change to ‘103 ‘The application manages large amounts of data from different information systems and shares them efficiently and securely in real-time’.

Line 105. Delete ‘Indeed’ and collapse paragraph

Line 105. Suggest change ‘differently’ to ‘separate’

Line 106. Suggest change ‘let’ to ‘allows’

Line 108. Change ‘makes sure the’ to ‘facilitates’

Line 125. Change ‘out-breaks’ to ‘outbreaks’

Line 127. Change ‘outbreaks’ to ‘outbreak’

Line 172. What is this? A law? ‘Through the DM 27 August 1994 n. 651 entered into force in Italy…

Line 176. ‘provides for control official serological test’ change to ‘demands serological testing’

Line 259. ‘near’ suggest change to ‘nearby’

Line 262. ‘let to establish’ replace with ‘establishing’

Line 264. ‘(three of which, were outbreaks)’. I’m a bit confused by this. Do you mean 3 out of the 4 farms had confirmed outbreaks?

Line 289. Is the whole herd tested at the same time? This has the effect of increasing the sensitivity of the test as it is effectively not a single animal test if the whole herd is restricted if a positive animal is found. This is an important consideration.

Line 310. ‘and outbreak management’ Does this refer to the field staff? Maybe rewrite to clarify.

Line 317. ‘racing’ suggest change to ‘racing against’

Line 325. ‘always explored’ change to ‘always be explored’

Line 333. ‘So having data about traded animals’ change to ‘Thus, having data on traded animals’

Line 353. ‘highlighted’ change to ‘highlights’

Reviewer #2: In this manuscript, the authors present the application T-Racing that they developed to support epidemiological investigations through interacting maps, graphs, and tables. This operational tool relies on three Italian sources of data: 1. the National Database for Animal Registration (BDN), 2. the National Animal Disease Notification System (SIMAN), and 3. the National Reference Centre for Whole Genome Sequencing of microbial pathogens database and bioinformatics analysis (GENPAT). The application relies on temporal network analysis for tracing back and tracing forward movements, related to an infected farm. The identified users for this application are healthcare professionals, epidemiologists, and relevant authorities. The manuscript describes the technical architecture of the application, shows a panel of screen shots of the application and describes study cases on bovine brucellosis and bovine tuberculosis. This is an interesting paper, well-written and valuable for the field.

Main comments

The manuscript does not provide sufficient information about the data and the methods. There is only an inventory of the sources of data and one sentence on the methodology. More details are needed to understand how the application manage and analyse the data to obtain outputs.

Regarding data, the authors should also discuss how the application complies with the General Data Protection Regulation (GDPR), given that various types of users are identified and will have access outputs (movements tracing of individual animals, genomic information) that are sensitive.

The authors mentioned that the T-Racing application is currently in testing phase but no information is provided about this pilot study (see the specific comment below, Line 138). Besides, I suggest discussing the main results of these tests (i.e. strengths and weaknesses of the application) and provide users’ feedback regarding the friendliness and relevance of the application, and how the results from the testing phase were used to improve the application.

The manuscript provides a short description of the functionalities of the tools (supported by several screen shots of the application). However, there is currently no information about how the application should be calibrated (species selection, time window, etc.) for tracing back and tracing forward movements depending on the diseases of interest.

Specific comments

Introduction

- Line 38: TRACES has become TRACES-NT (new technology) for few years: https://webgate.ec.europa.eu/IMSOC/tracesnt-help/Content/en/index.html.

- Line 45-60: Avoid 1-2 sentences paragraphs. All these sentences should be gathered in one paragraph about the interest of tracing disease spread for disease surveillance and control.

- Lines 61-63: Please expand this part about existing tools for animal tracing and explain what are the current limits of these tools or what features are missing (justifying the development and use of T-racing)?

- Lines 64-77: All these sentences should be gathered in one paragraph.

Materials and Methods

- Lines 85-90: Please provide a description of the data and datasets of these three databases (including spatial and temporal unit of the data and detailed information on livestock owners and producers).

- Line 97: Add references of previous studies using such methods for tracing disease spread. Describe how the temporal network analyses were conducted (including specifications about spatial and temporal units).

- Lines 100-109: All these sentences should be gathered in one paragraph.

- Line 110: Are the tools proposed in the T-Racing application available to all types of users? How does the application comply with the General Data Protection Regulation (GDPR)?

Results

- Line 138: The testing phase is an important step in the development of a tool to ensure that it meets the needs of users. What does the testing phase consist of? Please provide some details about the number and types of users (authorities, operational actors, etc.). Which indicators are used for testing the application and evaluating its performances?

- Line 151 (and 158): please clarify the sentence: “and the red colour highlights paths followed by animals, singularly identified, (which?) had gone out of the seed”

- Line 161: These examples seem to be based on real (not simulated) data. If so, replace “simulation Exercise” by “Case studies”. If simulated data were used, please describe this step (methods, data, etc.) in the Materials and Method part.

- Line 172 : Does DM mean Ministerial Decree ? If so, specify it on line 171 and use DM at each occurrence (e.g. line178). Otherwise, replace by “Ministerial Decree”.

- Line 187: In this case study on bovine brucellosis, first you used a period of 2 years for tracing movements, and then, line 198, a period of 399 days to reduce the number of movements. How did you select these values? Please also explain more broadly in the Discussion how the application should be calibrated (species selection, time window, etc.) for tracing back and tracing forward movements depending on the diseases of interest. Is the application more appropriate for specific types of diseases (like, for example, slow spreading diseases)?

- Line 198: What are those three levels?

- Line 204-205: “The reduced network, … focusing on … outbreaks within the network, …” Please, clarify what these two networks refer to?

- Line L209 : Write brucela abortus in italics.

- Line 214: The application enables search on “individually identified animals”. Such outputs are certainly sensitive information and I am expecting that this will be a barrier to the use of the tool. Can you please explain how the application deals with this issue?

- Figures 3, 4, 5 and 7 are too small to be readable.

- Line 254 : “examples”

- Figure 6: What does the black link mean?

- Line 278: What tool do you refer to in the sentence “we employed a tool…”

Discussion

- Line 305-307: This sentence is exactly the same as the first one of the Introduction!

- Line 312-322: This is not Discussion. I suggest moving this paragraph to the Introduction to underline the interest of One Health in disease surveillance, and highlight how tracing disease spread is critical for effective disease control.

- Figure 8: This figure would be better positioned in the Results part to demonstrate the relevance of T-Racing in preventing and detecting diffusion pathways versus the simple farm isolation, using a retrospective analysis of data. In Figure 8, please describe more specifically the detection and prevention of cases when T-racing is used versus without T-Racing (=surveillance program only). Letters of the alphabet could be used to distinguish the four farms and explain the chain of events regarding the disease transmission between farms (disease introduction and spread) and detection (through surveillance program and using T-Racing). Specify what is the “surveillance programme”: programmed surveillance (serology twice a year) or abortion notification? What do the yellow arrows mean? Besides, I suggest using a darker colour than yellow to be more visible on the figure.

- Lines 333-337: These sentences (at least the main idea) should be in the Introduction to explain the interest of developing and using the T-Racing tool versus getting an extraction of the movement network through a query of the BDN (National Database for Animal Registration)?

- L356-357: How is this sentence related to the topic of the paper? How the tool can foster collaboration between sectors?

- Line 367: What do you mean by “reliable data”? What criteria should be taken into account to consider that data is reliable?

Additional comments

- Can the tool be used to trace emerging diseases (so-called Disease X)?

- Can the tool be used by other countries with similar data and if so, how would the tool be made available?

Reviewer #3: Introduction: It's partially vague. As a veterinarian and epidemiologist, I understand the problem, but what is state-of-the-art, and why is your tool unique? What motivated you to create this tool? Please also remove redundancy in several phrases. Be more explicit about the objectives of the work.

Methods: It's vague. The methodology (the tool) is the heart of your work; if we need to help to understand your tool clearly, how would we know its usefulness? Please restructure this part into sections and subsections. For example, the section on the tool should have an explanation of each module. The section on your experiments should provide details of the disease models and regions. Check for inspiration and organisation of the work via other methods/tools papers published in Plosone. For example: https://journals.plos.org/plosone/article?id=10.1371/journal.pone.0199960

Results: There is a mix between methods and results. Calrity and reorganisation is necessary.

Discussion: Its an introduction and not a discussion. I dont see how the results are original compared to simillar works and tools in Europe. Their originality, limitations and perspectives. This section needs most reworking.

Other questions:

Is the code of the R shiny app available?

How the tool was tested with end-users and co-created with end-users?

Why is the tool a demo and not operational?

The tool allows us to assess the risk based on animal movements only. What about human movements and contaminated products? For example, how would it be for African swine fever?

You mention the tool is important from a public health perspective. How do you envisage using it with human data and human outbreaks of Brucellosis and Tuberculosis, for example?

How do you envisage using the tool for movements originating outside Italy? Origin EU and outside EU, for example?

In the introduction and discussion, you discuss the importance of your tool for One Health; this term is overused because I would like to see more explicitly how the tool adds to the human and environmental Health presented in the methods and the results. Say it clearly if the tool is just for animal health and the perspectives are to enlarge to One Health.

Reviewer #4: The submitted manuscript aims to present T-Racing, a tool to perform contact tracing analysis in case of animal diseases outbreaks in a timely and precise way. Having to deal with similar problems in my day-to-day work, I think tool like this (which other than animal movements, in includes data such as the pathogen sequence and the type of farms) are fit for purpose and could help improve the speed of such analyses. However, the manuscript presents two major issues. First, it is very poorly written, and it needs a substantial restructuring (see further comments). Second, it is hard to evaluate the tool without having access to it. I would suggest the author to resubmit the manuscript once a demo of the tool is available online (or provide it) and with a dummy dataset (i.e. made-up network/farms/outbreaks) to allow running the scripts.

Comments:

1. The “Introduction” section is poorly written, it is a collection of statements without a logic flow. Please rewrite.

2. Lines 94-96: please report how the tool complies with the privacy laws (GDPR, etc).

3. Lines 97-98: “Temporal Network Techniques” is too generic, please say which ones and provide references.

4. Table 1: in my understanding R packages sp and rgdal might not going to be maintained anymore, their functionalities could be found in sf and terra. I’d suggest to investigate this issue and eventually change the code.

5. Why does Fig 2 not show a point in the Marche region where the brucellosis outbreak was detected? If it is because the outbreak was traced back between Sicily and Tuscany? If this is the case, please add some indication on where the outbreak was found (Marche?), otherwise it is very confusing.

6. In general, figures are not consistent with the text and look randomly placed.

7. Paragraph 3.1.1 (and 3.1.2): the first describing general information about brucellosis (and bTB) should go into introduction or material and methods.

8. Lines 196-199: it is not clear how you narrowed down from 386 to 38 farms (and 1826 to 199 movements). Which type of tracing was done? Is it going to be available in T-Racing?

9. Lines 278-280: similarly, which tool was used to connect “outbreaks”? With which criteria movements/farms were excluded?

6. PLOS authors have the option to publish the peer review history of their article (what does this mean? ). If published, this will include your full peer review and any attached files.

**Do you want your identity to be public for this peer review?** For information about this choice, including consent withdrawal, please see our Privacy Policy .

Reviewer #1: No

Reviewer #2: No

Reviewer #3: **Yes: ** Elena Arsevska

Reviewer #4: No

---

## [Author Response · Author response to Decision Letter 1]

30 Jun 2024

We attached the doc file with answers to reviewers

---

## [Decision Letter · Decision Letter 1]

8 Aug 2024

PONE-D-24-01354R1T-Racing: a modern tool for supporting epidemiological investigation in animal disease outbreaksPLOS ONE

Dear Dr. Candeloro,

Thank you for submitting your manuscript to PLOS ONE. After careful consideration, we feel that it has merit but does not fully meet PLOS ONE’s publication criteria as it currently stands. Therefore, we invite you to submit a revised version of the manuscript that addresses the points raised during the review process.

We look forward to receiving your revised manuscript.

Kind regards,

Anna Bernasconi, PhD

Academic Editor

PLOS ONE

**Additional Editor Comments:**

Dear authors, thank you for providing the revised manuscript. Please see the comments of three reviewers. There are still important requests to be addressed. We will be waiting for your revised work to assess it again.

Reviewers' comments:

Reviewer's Responses to Questions

**Comments to the Author**

1. If the authors have adequately addressed your comments raised in a previous round of review and you feel that this manuscript is now acceptable for publication, you may indicate that here to bypass the “Comments to the Author” section, enter your conflict of interest statement in the “Confidential to Editor” section, and submit your "Accept" recommendation.

Reviewer #1: (No Response)

Reviewer #3: (No Response)

Reviewer #5: (No Response)

2. Is the manuscript technically sound, and do the data support the conclusions?

Reviewer #1: Yes

Reviewer #3: Partly

Reviewer #5: Yes

3. Has the statistical analysis been performed appropriately and rigorously? 

Reviewer #1: N/A

Reviewer #3: N/A

Reviewer #5: Yes

4. Have the authors made all data underlying the findings in their manuscript fully available?

Reviewer #1: Yes

Reviewer #3: No

Reviewer #5: Yes

5. Is the manuscript presented in an intelligible fashion and written in standard English?

Reviewer #1: Yes

Reviewer #3: Yes

Reviewer #5: Yes

6. Review Comments to the Author

**Reviewer #1:**  Dear authors,

Thank you for addressing the concerns and observations I made in the initial review of this paper. In my opinion, you have greatly improved the manuscript. I reiterate on my past comment that this work isn't unique to Italy. Many other countries build their own tracing tools. However, this is usually done by IT professionals under contract and rarely gets shared to the wider scientific community which is why I'm pleased you have gone to the effort to seek to publish your work. As this is not a conventional research paper, it doesn't quite fit the template for presenting your methodologies and results. You have improved the structure and layout in this version of the manuscript and I've added some more minor edit suggestions below:

Line 20: delete ‘Therefore’

Line 21: replace “and almost in real-time” with “in near real-time”

Line 35: delete “emergency”

Line 45,46. Delete “On the other hand”.

Line 212: Replace “In the end, such kind of strategies” with “Ultimately, such strategies”

Line 213: Delete “and” from “(and thus)

Line 233: Add “The”… “The T-Racing interface”

Line 234: “graphs, maps,”

Section 2.3 is poorly laid out in terms of flow of information. I would suggest a adopting clearer headings and subheadings rather than multiple descriptions followed by bullet points.

Line 255: Replace “make” with “select”

Section 2.6 Demo and 3. Results.

https://demo.izs.it/app_direct/tracing/

I can access the interface but can’t select any dummy data.

Line 503: “inform the response in near real-time and to allow”

Line 427: Replace “focus” with “focused”

Line 533: I would suggest changing “Complementarily,” to “Coupling”.

Lines 533 – 550. This is mostly repetition and overly wordy. Is it necessary or could it be replaced with a more simple and humble sentence?

**Reviewer #3: ** Dear authors,

Thank you for providing an improved version of the manuscript.

I provide my detailed comments in the reviwed manuscript attached.

I have several general suggestions:

- you should clearly mention that the tool is not open source and freely available and say why

- if the tool is generic should be underlined as an advantage and say how it can be of use one day for other diseases and other services in Italy, for eample One health, as you mentuon this in the discussion

- you dont discuss the limitations of the tool - this should be clearly addressed in the discussion

Kind regards

**Reviewer #5:**  The subject of the article is of interest.

I have revisions to suggest to authors and some questions, see below:

Introduction section

Line 47: specify that 2.1 million people is worldwide (I suppose it is that).

Line 54-60: Is it really relevant to have this description of TRACES-NT because this databasis is not used in the article? If it is maintained, it should be specified that it is only accessible to competent authority and considered as sensitive data.

Material and method

• I was surprised that no wildlife data (wildlife TB and brucellosis positive cases) were considered for TB and brucellosis. Are these data not available? Is it a possibility to improve the tool for future? It would have also justified the “one health” approach cited in the article.

• How rendering plants and slaughterhouses are considered in the tool, are they considered in the same way as a farm in the network analysis? Idem for transit section that are cited in the result section (line 381) but we don’t have information on how it is considered in the tool in the material and method section.

• Line 242-243: the tool considere spatial buffer using distance by kilometers. Do you think of using pasture data to improve your tool in the future?

• Line 330-331 and 340: how these parameters were determined? Taking into account movements only over the past 13 months seems for instance restrictive considering TB, a chronic infection that can be difficult to detect. Was it based on experts’ elicitation?

• Line 350: considering TB as important zoonosis seems not adequate. It has important economic impact due to movement and commercial restrictions for farmers but in Europe TB human cases are not anymore due to bovine tuberculosis as far as I know. Is there any publication that can support this zoonotic aspect in European countries? It seems to me that it is not a big issue for human health anymore.

Results

• Line 471: it seems here that pasture data are included but I can’t find this information in the material and method section.

Discussion

Several times in the article, and especially in the discussion section, the “One health” approach is cited but I can’t see how the tool could be considered as a One health approach tool at that stage. There is non-human health data which could be relevant only for brucellosis. For BT it is relevant to consider only human cases due to bovine tuberculosis due to contamination in Italy which I suppose is very rare, nearly no case no? If data on human cases due to bovine tuberculosis and brucellosis are available it would be great to add it in the introduction section. If BT is of importance on a zoonotic aspect, authors should consider to improve the tool in the future, to add information on farm production type so as to distinguish raw milk production farms.

At that stage there is no data on wildlife surveillance considered which could have justified the one health approach with the environmental sector.

7. PLOS authors have the option to publish the peer review history of their article (what does this mean? ). If published, this will include your full peer review and any attached files.

**Do you want your identity to be public for this peer review?** For information about this choice, including consent withdrawal, please see our Privacy Policy .

Reviewer #1: No

Reviewer #3: No

Reviewer #5: **Yes: ** Céline Dupuy

---

## [Author Response · Author response to Decision Letter 2]

10 Sep 2024

Please, find responses to reviewer in the attached word file "T-Racing-Response to reviewers comments.doc"

---

## [Decision Letter · Decision Letter 2]

27 Sep 2024

PONE-D-24-01354R2T-Racing: a modern tool for supporting epidemiological investigation in animal disease outbreaks in ItalyPLOS ONE

Dear Dr. Candeloro,

Thank you for submitting your manuscript to PLOS ONE. After careful consideration, we feel that it has merit but does not fully meet PLOS ONE’s publication criteria as it currently stands. Therefore, we invite you to submit a revised version of the manuscript that addresses the points raised during the review process.

We look forward to receiving your revised manuscript.

Kind regards,

Anna Bernasconi, PhD

Academic Editor

PLOS ONE

Journal Requirements:

**Additional Editor Comments:**

Dear authors,

please take into consideration the comments of R1 and R3 that require a minor revision of your work.

Please note that the review of R1 was performed in two different points in time. Indeed, the reviewer, initially, was not able to access the application. Later, he/she was able to access it and provided another comment (below, you find both reviews).

Reviewers' comments:

Reviewer's Responses to Questions

**Comments to the Author**

1. If the authors have adequately addressed your comments raised in a previous round of review and you feel that this manuscript is now acceptable for publication, you may indicate that here to bypass the “Comments to the Author” section, enter your conflict of interest statement in the “Confidential to Editor” section, and submit your "Accept" recommendation.

Reviewer #1: (No Response)

Reviewer #3: All comments have been addressed

2. Is the manuscript technically sound, and do the data support the conclusions?

Reviewer #1: Yes

Reviewer #3: Yes

3. Has the statistical analysis been performed appropriately and rigorously? 

Reviewer #1: N/A

Reviewer #3: Yes

4. Have the authors made all data underlying the findings in their manuscript fully available?

Reviewer #1: Yes

Reviewer #3: Yes

5. Is the manuscript presented in an intelligible fashion and written in standard English?

Reviewer #1: Yes

Reviewer #3: Yes

6. Review Comments to the Author

Reviewer #1: Dear authors

Thank you for updating the manuscript to account for the last round of comments!

My main issue at present is I can't access the demonstration. The link in the paper isn't currently working: https://demo.izs.it/app_direct/tracing/ Could you please resolve this so I can see everything working?

There is still a bit of repetition in the introduction and discussion (and the start of the materials and methods section). You could remove some of this without impacting on the message you're conveying.

Finally, some minor editorial comments:

Line 60: This has been pasted twice - remove ". t is only accessible to

61 competent authorities and is considered sensitive data."

Line 71-73: "When zoonoses are considered, this also means protecting public health, once contact tracing has the subsequent effect of providing assessments to stakeholders which is useful for quickly identifying and implementing containment, surveillance, and control systems." Rephrase as "once contact tracing has the subsequent effect" does not make sense.

Line 85: Delete 'While'

Line 91: "Currently, due to the COVID-19 global pandemic" Rephrase, "Due to the recent COVID-19 global pandemic"

Line 99: "it lacks of integration of heterogeneous" Rephrase "it lacks the ability to intergrate heterogeneous..."

Line 102-103: "but its complexity need for specialized expertise make it less accessible" Rephrase "but its complexity requires specialized expertise making it less accessible..."

Line 233: Delete "Nowadays, "

Line 349: "The following case studies were considered:" Consider rephrasing to somthing like "To demonstrate the real-world utility of the application the following case studies were considered:"

========

REVIEWER 1  comment provided at later stage

"Feedback to the authors. I would make the following suggestions:

The interactive demonstration of the application is very useful and the readers should probably be made more aware of its exitance, perhaps by making reference to it in the introduction and abstract?

The legend isn't obvious. It took me some time to find the legend and I couldn't re-size it. Would it be possible to have the legend in English on the demonstration version?

When selecting nodes on the map, I couldn't find a way to de-select them.

For some reason, the application didn't work on Google Chrome (PC, Win 11) but did work on Edge (PC Win 11)"

Reviewer #3: Dear authors,

Thank you for sending an improved manuscript version and considering my comments carefully.

My main comment for this round of revision is that you present the tool as an OH tool and, in the beginning, talk about its usefulness to PH agencies. The tool is currently being developed in the AH and used internally for IZS Teramo. That is fine and should be acknowledged from the start.

Check the article where we present a platform developed for a specific purpose (AH in France in our case): https://journals.plos.org/plosone/article?id=10.1371/journal.pone.0199960

So, I disagree that the tool can currently be useful for PH practitioners unless you mention in the discussion how you share information between ISZ Teramo and ISS Italy, for example, and how you communicate the results of the investigation, etc., to your peers for efficient risk assessment and decision-making (Oh approach)—or other examples of how PH agencies in Italy (e.g., ISS Roma) use this tool currently.

Otherwise, please improve the discussion of how the tool can evolve into an OH tool and provide suggestions on collaboration and sharing information with the PH sectors, for example. This is not a limitation but a perspective and a vision I am sure you have for the tool. So please don’t hesitate to be open and share it with the readers.

With best regards,

I hope you will find my comments constructive.

7. PLOS authors have the option to publish the peer review history of their article (what does this mean? ). If published, this will include your full peer review and any attached files.

**Do you want your identity to be public for this peer review?** For information about this choice, including consent withdrawal, please see our Privacy Policy .

Reviewer #1: No

Reviewer #3: **Yes: ** Elena Arsevska

---

## [Author Response · Author response to Decision Letter 3]

11 Nov 2024

PONE-D-24-01354R2

T-Racing: a modern tool for supporting epidemiological investigation in animal disease outbreaks in Italy

PLOS ONE

Reviewer #1: Dear authors

Thank you for updating the manuscript to account for the last round of comments!

My main issue at present is I can't access the demonstration. The link in the paper isn't currently working: https://demo.izs.it/app_direct/tracing/ Could you please resolve this so I can see everything working?

Thank you for your message. We're glad to hear that you were able to access the demonstration successfully on your own.

There is still a bit of repetition in the introduction and discussion (and the start of the materials and methods section). You could remove some of this without impacting on the message you're conveying.

Answer: Thank you for your feedback regarding the repetition in the introduction, discussion, and the beginning of the materials and methods section. In response to your comments, we have carefully reviewed the text and identified overlapping content between the introduction, M&M, and discussion. We have removed certain redundant phrases and concepts without compromising the overall message we aim to convey. This has helped to enhance the flow and coherence of the manuscript. We are grateful for your guidance, which has contributed to refining our work.

Finally, some minor editorial comments:

Line 60: This has been pasted twice - remove ". t is only accessible to

61 competent authorities and is considered sensitive data."

Answer: the test has been deleted

Line 71-73: "When zoonoses are considered, this also means protecting public health, once contact tracing has the subsequent effect of providing assessments to stakeholders which is useful for quickly identifying and implementing containment, surveillance, and control systems." Rephrase as "once contact tracing has the subsequent effect" does not make sense.

Answer: Thank you for your insightful comment. We have revised the sentence regarding contact tracing for clarity. The updated version now emphasizes its pivotal role in identifying necessary actions to break the transmission chain and protect public health.

Line 85: Delete 'While'

Answer: Done

Line 91: "Currently, due to the COVID-19 global pandemic" Rephrase, "Due to the recent COVID-19 global pandemic"

Answer: Done

Line 99: "it lacks of integration of heterogeneous" Rephrase "it lacks the ability to intergrate heterogeneous..."

Answer: Done

Line 102-103: "but its complexity need for specialized expertise make it less accessible" Rephrase "but its complexity requires specialized expertise making it less accessible..."

Answer: Done

Line 233: Delete "Nowadays, "

Answer: Done

Line 349: "The following case studies were considered:" Consider rephrasing to somthing like "To demonstrate the real-world utility of the application the following case studies were considered:"

Answer: Done

========

REVIEWER 1  comment provided at later stage

"Feedback to the authors. I would make the following suggestions:

The interactive demonstration of the application is very useful and the readers should probably be made more aware of its exitance, perhaps by making reference to it in the introduction and abstract?

Answer: We have now referenced the interactive demo of T-Racing in both the abstract and introduction sections.

The legend isn't obvious. It took me some time to find the legend and I couldn't re-size it. Would it be possible to have the legend in English on the demonstration version?

Answer: Thank you for your feedback regarding the legend. One of the improvements we plan to implement is making the legend dynamic so that it remains aligned with the graph content instead of being static as it is now, which will enhance its visibility. In the meantime, we have updated the legend images, translated them into English as requested, and these are now available in the demonstration version of the application.

When selecting nodes on the map, I couldn't find a way to de-select them.

Answer: Thank you for your observation. To deselect a node on the map, simply click anywhere outside the graph in the white space of the right-hand view. This will clear the selection.

For some reason, the application didn't work on Google Chrome (PC, Win 11) but did work on Edge (PC Win 11)"

Answer: Thank you for bringing this issue to our attention. We have successfully verified that the application runs on Google Chrome (PC, Win 11), Version 129.0.6668.70 (64-bit). Please ensure that your version of Google Chrome is updated.

Reviewer #3: Dear authors,

Thank you for sending an improved manuscript version and considering my comments carefully.

My main comment for this round of revision is that you present the tool as an OH tool and, in the beginning, talk about its usefulness to PH agencies. The tool is currently being developed in the AH and used internally for IZS Teramo. That is fine and should be acknowledged from the start.

Check the article where we present a platform developed for a specific purpose (AH in France in our case): https://journals.plos.org/plosone/article?id=10.1371/journal.pone.0199960

So, I disagree that the tool can currently be useful for PH practitioners unless you mention in the discussion how you share information between ISZ Teramo and ISS Italy, for example, and how you communicate the results of the investigation, etc., to your peers for efficient risk assessment and decision-making (Oh approach)—or other examples of how PH agencies in Italy (e.g., ISS Roma) use this tool currently.

Otherwise, please improve the discussion of how the tool can evolve into an OH tool and provide suggestions on collaboration and sharing information with the PH sectors, for example. This is not a limitation but a perspective and a vision I am sure you have for the tool. So please don’t hesitate to be open and share it with the readers.

With best regards,

I hope you will find my comments constructive.

Thank you for your valuable feedback and for highlighting these important points. We acknowledge that the T-Racing tool is currently used in the context of animal health (AH). However, it is important to emphasize that in Italy, both animal health (AH) and public health (PH) services operate under the Ministry of Health, differently from how this occurs in other countries. This structural integration facilitates a closer collaboration between these sectors, particularly in the context of zoonotic disease management and environmental disaster responses, making it easier to adopt a One Health approach. This is particularly relevant for the Italian system, where PH and AH authorities regularly collaborate during outbreaks, and share information effectively.

Regarding your comment on how T-Racing is currently used, while the tool is still primarily focused on AH data, its integration within the national VetInfo system ensures that it will be accessible to both AH and PH agencies, enabling them to collaborate closely in risk assessment, outbreak investigation, and decision-making processes. We believe that this structural collaboration between public and animal health authorities in Italy—unique to our national system—already aligns with the One Health framework, as it inherently supports interdisciplinary cooperation.

In the future, as suggested, we plan to further develop T-Racing to include more PH data, enhancing its utility for public health practitioners and strengthening its role as a One Health tool. This will allow for the sharing of more comprehensive information between institutes and stakeholders in both the AH and PH sectors. We have expanded the discussion to reflect this vision, highlighting the tool’s potential evolution and future contributions to the One Health approach. An example of the value and efficacy of this collaboration can be found in the following paper:

Sorgentone et al., A large foodborne outbreak of campylobacteriosis in kindergartens and primary schools of Pescara, Italy, May-June 2018. 2021 2021 Journal of Medical Microbiology, 70(3). https://doi.org/10.1099/jmm.0.001262 Epub 2021 Jan 20.

The remaining comments from Reviewer 3 have been addressed directly in the manuscript file in the track changes version. Thank you again for your constructive feedback, which has greatly helped us improve the clarity and direction of our manuscript.

---

## [Decision Letter · Decision Letter 3]

26 Dec 2024

T-Racing: a modern tool for supporting epidemiological investigation in animal disease outbreaks in Italy

PONE-D-24-01354R3

Dear Dr. Candeloro,

We’re pleased to inform you that your manuscript has been judged scientifically suitable for publication and will be formally accepted for publication once it meets all outstanding technical requirements.

Kind regards,

Anna Bernasconi, PhD

Academic Editor

PLOS ONE

Additional Editor Comments (optional):

Dear Authors, I am happy to accept your manuscript in the current form, after several rounds of revision that improved the content and made its suitable for this journal.

Reviewers' comments:

Reviewer's Responses to Questions

**Comments to the Author**

1. If the authors have adequately addressed your comments raised in a previous round of review and you feel that this manuscript is now acceptable for publication, you may indicate that here to bypass the “Comments to the Author” section, enter your conflict of interest statement in the “Confidential to Editor” section, and submit your "Accept" recommendation.

Reviewer #1: All comments have been addressed

2. Is the manuscript technically sound, and do the data support the conclusions?

Reviewer #1: Yes

3. Has the statistical analysis been performed appropriately and rigorously? 

Reviewer #1: N/A

4. Have the authors made all data underlying the findings in their manuscript fully available?

Reviewer #1: Yes

5. Is the manuscript presented in an intelligible fashion and written in standard English?

Reviewer #1: Yes

6. Review Comments to the Author

Reviewer #1: Dear Editor and authors,

Thank you for your continued efforts to bring this work to publication. I'm happy to accept the manuscript in it's current form.

7. PLOS authors have the option to publish the peer review history of their article (what does this mean? ). If published, this will include your full peer review and any attached files.

**Do you want your identity to be public for this peer review?** For information about this choice, including consent withdrawal, please see our Privacy Policy .

Reviewer #1: No

---

## [Editor Report · Acceptance letter]

PONE-D-24-01354R3

PLOS ONE

Dear Dr. Candeloro,

I'm pleased to inform you that your manuscript has been deemed suitable for publication in PLOS ONE. Congratulations! Your manuscript is now being handed over to our production team.

Kind regards,

on behalf of

Dr. Anna Bernasconi

Academic Editor

PLOS ONE